# Alternative Methods to Animal Testing for the Safety Evaluation of Cosmetic Ingredients: An Overview

**Maria Pilar Vinardell \*** iD **and Montserrat Mitjans** iD

Departament of Biochemistry and Physiology, Faculty of Pharmacy and Food Sciences, Universitat de Barcelona, Av. Joan XXIII 27-31, Barcelona 08028, Spain; montsemitjans@ub.edu
**\*** Correspondence: mpvinardellmh@ub.edu; Tel.: +34-934-024-505

**Abstract:** The safety of cosmetics sold in Europe is based on the safety evaluation of each individual ingredient conducted by those responsible for putting the product on the market. However, those substances for which some concern exists with respect to human health (e.g., colorants, preservatives, UV-filters, nanomaterials) are evaluated at the European Commission level by a scientific committee, currently called the Scientific Committee on Consumer Safety (SCCS). According to the Cosmetics Regulation (European Commission, 2009), it is prohibited in the European Union (EU) to market cosmetic products and ingredients that have been tested on animals. However, the results of studies performed before the ban continue to be accepted. In the current study, we evaluated the use of in vitro methods in the dossiers submitted to the SCCS in the period between 2013 and 2016 based on the published reports issued by the scientific committee, which provides a scientific opinion on these dossiers. The results of this evaluation were compared with those of an evaluation conducted four years previously. We found that, despite a slight increase in the number of studies performed in vitro, the majority of studies submitted to the SCCS is still done principally in vivo and correspond to studies performed before the ban.

**Keywords:** alternative methods; cosmetic ingredients; safety evaluation; animal ban; dossiers; SCCS; in vitro; in vivo

## 1. Introduction

The safety evaluation of cosmetics in Europe is based on the evaluation of each individual ingredient. Article 3 of the Cosmetics Regulation specifies that a cosmetic product made available on the market shall be safe for human health when used under normal or reasonably foreseeable conditions of use. In practice, cosmetic products have rarely been associated with serious health hazards, which, however, does not mean that cosmetics are safe to use per se. Particular attention needs to be paid to the long-term safety aspects, since cosmetic products may be used extensively over much of the human lifespan and sensitive groups of the population such as children, old people, and pregnant women may be involved. Therefore, the safety-in-use of cosmetic products has been established in Europe by controlling the substances, their chemical structures, toxicity profiles, and exposure patterns.

For those substances for which some concern exists with respect to human health (e.g., colorants, preservatives, UV-filters), the safety evaluation is conducted at the European Commission (EC) level by the Scientific Committee on Consumer Safety (SCCS).

The SCCS was established in 2008 to replace the former Scientific Committee of Consumer Products (SCCP). Before 1997, the opinions adopted by the Scientific Committee on Cosmetology at the Commission's request were included in EC-Reports. Between 1997 and 2004, all Scientific Committee opinions were published on the Internet and can be accessed through the relevant Committee's Website. All SCCS opinions can be located via the ingredient's category and the adoption

date [1]. Since November 2015, the opinions have also been published in Regulatory Toxicology and Pharmacology.

One of the responsibilities of the SCCS is to recommend a set of guidelines to be taken into consideration by the cosmetic and raw material industry when developing studies to be used in the safety evaluation of cosmetic substances. The SCCS evaluates the dossiers submitted by industry through the Directorate General of Health and Food Safety (DG SANTE). The ingredients evaluated by the SCCS correspond to those in the Annexes of Regulation; more specifically, substances with restrictions in Annex III, and colorants, preservatives and UV-filters in Annexes IV, V and VI, respectively.

The determination of the toxic potential of a cosmetic substance is based on a series of toxicity studies and forms part of the hazard identification. Alternative methods to animal testing of cosmetic ingredients have not been mandatory in Europe since March 2013, according to the Commission. Traditionally, toxicological data relevant for man have been obtained by investigating the toxicological profiles of the substances on animals, if possible using the same exposure route as in humans (topical, oral or inhalation route). Toxicological studies are often performed by the oral route and then the corresponding extrapolation to the dermal route should be done.

When the dossier of a cosmetic substance is submitted for evaluation by the SCCS, the manufacturer should provide the Commission with information about acute toxicity (if available); irritation and corrosivity to skin and eye; skin sensitization; dermal/percutaneous absorption; repeated dose toxicity; mutagenicity/genotoxicity; carcinogenicity; reproductive toxicity; toxicokinetics; photo-induced toxicity and human data as detailed in the notes of guidance of the SCCS revised in 2016 [2].

According to the Cosmetics Regulation (European Commission, 2009), it is prohibited in the EU to market any cosmetic products and ingredients that have been tested on animals for most of their human health effects, including acute toxicity. This means that the cosmetic industry needs to have alternative approaches available to test the safety of ingredients of consumer products.

The use of animals in cosmetic testing was forbidden from March 2009, onwards, with the exception of studies on repeat dose, which were permitted until March 2013. From this date, new studies of old or new substances intended for cosmetic were required to be conducted without animals.

In a previous paper, we studied the use of alternative methods in the dossiers submitted to the SCCS in the period April 2009–March 2013, immediately prior to the ban. In the current study, we compare the use of alternative methods in studies presented in the dossiers before and after the complete ban in March 2013.

## 2. Materials and Methods

The study material consisted of SCCS opinions issued between April 2013 and March 2016 after the nomination of SCCS new members. No confidential data were used, as all information came from opinions downloaded from the Committee's website. In the present study, only the full opinions were considered; we did not take into account addenda or special opinions on a particular item, such as sensitization.

Each opinion was analyzed for each different methodology in the toxicological section, recording the procedure used and distinguishing between procedures based on in vivo or in vitro models. The percentage of studies performed using non-animal models was compared to that of studies using animal models and in some cases with human data.

A total of 41 dossiers were analyzed, 19 corresponding to hair dyes and 22 to other substances including UV filters, fragrances, and preservatives, among other ingredients. The results obtained were compared with those of the opinions published in the period 2009–2013 and other previously published results [1].

## 3. Results and Discussion

The SCCS opinions are currently organized into those for hair dyes, cosmetic ingredients and nanomaterials, but, in the period evaluated in the previous paper [1], the opinions were organized into those for fragrances, hair dyes, preservatives, UV-filters and other substances. In the current paper, for comparative purposes, we combined all these categories. There were fewer SCCS opinions in the period 2013−2016 than the previous period, as the later period was shorter, so, for comparative purposes, we used percentage.

Studies performed with animals were considered acceptable if they were done before the ban on animal use. The requirements for the use of in vivo and in vitro methods in the safety evaluation of cosmetic ingredients performed by the SCCS are that the methods follow a guideline and were conducted based on the principle of good laboratory practices (GLP). If a study does not satisfy these criteria, then it is not considered by the SCCS and new data are required.

We have recorded the total number of studies performed in vivo, in vitro or in humans. Some ingredients have been studied with different methodologies for a specific toxicological study.

### 3.1. Acute Toxicity

Studies of acute toxicity are not always necessary in the dossiers submitted to the SCCS but were usually present in the dossiers supplied by industry, and, in most cases, were performed in rats, mice and rabbits. The oral route was the most commonly employed route, but dermal administration was also used, and, in a few cases, information about the inhalation route was also supplied. All accepted methods for determining acute oral toxicity are based on in vivo experiments after the single administration of a few doses and estimation of the dose causing the death in 50% of the animals. The 3T3 Neutral Red Uptake cytotoxicity assay has been proposed to identify substances not requiring classification as acute oral toxicants under EU regulations [3], but none of the dossiers evaluated in the period 2013–2016 or 2009–2013 used this methodology.

### 3.2. Eye Irritation

In the dossiers submitted to the SCCS, almost all studies were performed on albino rabbits and only a few used in vitro methods. However, there has been a slight increase in the use of in vitro methods. The majority of the in vivo studies were performed on rabbits and followed the the Organisation for Economic Co-operation and Development (OECD) 405 guideline, which was adopted in 1981 and updated successively in 1987, 2002, and recently in 2012 [4]. Figure 1 shows the percentage of studies performed to evaluate eye irritation in the two periods studied. It shows that there has been a decrease in the number of studies performed in animals and in human volunteers in recent years. In the case of human volunteers, the SCCS always indicates that it considers such studies unethical, which could have led to the observed reduction.

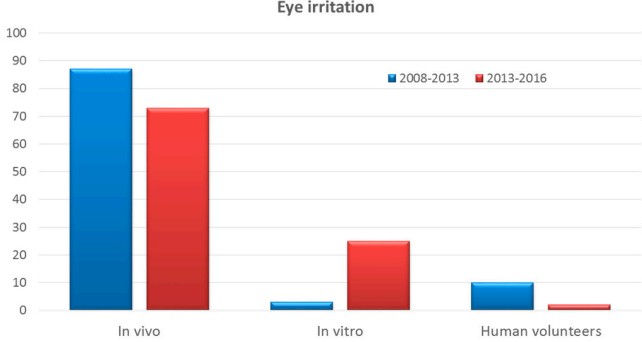

**Figure 1.** Evolution of the percentage of studies performed in vivo, in vitro and in human volunteers to evaluate the potential eye irritation of cosmetic ingredients-number of studies 82 (2008–2013) and 45 (2013–2016).

The study of eye irritation is one of the more classical studies performed on animals, usually rabbits and was developed many years ago to evaluate cosmetics and cosmetic ingredients [5]. This method is highly controversial and much efforts have been made to develop alternative methods [6].

Among the in vitro methods used in the dossiers of different ingredients presented to the SCCS for evaluation, there is the isolated chicken eye (ICE) and the Bovine Corneal Opacity and Permeability Test (BCOP), two validated methods appearing in the respective OECD guidelines [7,8]. Another method is the Het-Cam, a non-validated method that is often used in the cosmetic industry due to its cheap cost [9].

Only one study was performed in ten human volunteers, involving an eyelash-waving product containing thioglycolic acid. The human study did not conform to Good Clinical Practice (GCP), but the overall requirements of GCP were fulfilled by the trial.

A method that uses reconstructed human cornea-like epithelium (RhCE), which closely mimics the histological, morphological, biochemical and physiological properties of the human corneal epithelium, has recently been included in the OECD guidelines. This test guideline describes an in vitro procedure allowing the identification of chemicals (substances and mixtures) not requiring classification and labelling for eye irritation or serious eye damage in accordance with UN Globally Harmonized System of Classification and Labeling of Chemicals (UN GHS) [10]. Only two of the studies presented to the SCCS used this methodology with one of the commercial three-dimensional RhCE tissue constructs proposed by the guideline.

The in vitro studies included in the dossiers were performed after 2013, the year when the ban on animal testing entered into force.

There is a need for alternative approaches to replace the in vivo rabbit Draize eye test for evaluation of eye irritation of cosmetic ingredients given the animal ban and the potential contact with the eyes of cosmetics designed for application around the eyes or formulations to be applied on the head and accidentally contacted with eyes. Several assays have been developed, some of which have undergone formal validation, but no single in vitro assay has been validated as a full replacement for the rabbit Draize eye test. Each of the in vitro assays is related to a specific endpoint of ocular irritation and gives only partial information on the mode of action of the material tested. Thus, the weight-of-evidence (WoE) approach and results of multiple selected in vitro tests are needed to accurately estimate the degree of eye irritation caused by cosmetic ingredients [9,11].

In the development of alternative methods, it is crucial to know which of the effects assessed in the in vivo Draize eye test are responsible for driving the UN GHS and European Classification, Labelling and Packaging (EU CLP) classification systems and the influence on the predictive capacity of new in vitro methods. A number of key criteria should be taken into consideration when selecting reference chemicals for the development, evaluation and/or validation of alternative methods and/or strategies for serious eye damage/eye irritation testing [12]. Cosmetics Europe recently compiled a database of Draize data used for past validation activities. An evaluation of the various in vivo drivers of classification compiled in the database was performed to establish which of these were most important from a regulatory point of view, and, from the results obtained, they suggested the need for a critical revision of the UN GHS/EU CLP decision criteria for the Cat 1 classification of chemicals [13].

### 3.3. Skin Irritation

The accepted method of testing skin irritation was adopted in 1981 and updated in 2002 [14]. The method is based on the use of rabbits, but other species such as guinea pig and mouse have been used to a lesser extent for the evaluation of cosmetic ingredients.

As in the case of eye irritation, there has been a decrease in the number of studies performed in vivo and in human volunteers in recent years. The percentage of studies performed in vitro increased from 5% to 20% between the two study periods and the percentage of studies on human volunteers decreased (Figure 2).

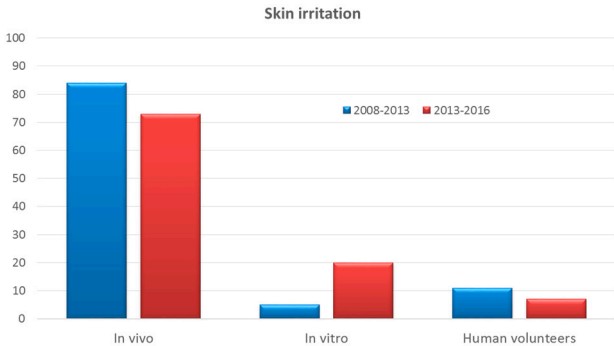

**Figure 2.** Evolution of the percentage of studies performed in vivo, in vitro and in human volunteers to evaluated potential skin irritation of cosmetic ingredients—number of studies 79 (2008–2013) and 44 (2013–2016).

One of the in vitro methods used to test skin irritation is the rat skin Transcutaneous Electrical Resistance test (TER), which is really an ex vivo method, because a laboratory animal (rat) is required to obtain the skin with subsequent determination of the TER. This method was not considered in the present study as an alternative method to animal testing and was not recorded in this context. The method was adopted in 2004 by the OECD guidelines and updated in 2013 [15], but really it determines skin corrosion rather than skin irritation.

The alternative method used in the studies for SCCS evaluation is the one based on reconstructed human epidermis. The method was accepted in the OECD guidelines as OECD439 in 2010 and was updated in 2013 and then in 2015 [16]. Taking the Episkin™ method as an example, the SCCS expressed concerns with regard to the potential interference with colour formation by reducing substances, hair dyes and colorants, and the SCCS expressed the opinion that the modified Episkin™ method did not provide sufficient proof that the 3-(4,5)-dimethyl-2-thiazolyl-2,5-diphenyl-2H-tetrazolium bromide (MTT) test could be used as a suitable endpoint to test color ingredients/hair dye substances for their potential skin irritant properties. A different endpoint, not involving optical density quantification, should be envisaged [17]. A recent study recommended that HPLC/UPLC-spectrophotometry to measure formazan should be included in the list of in vitro methods based on reconstructed human epidermis, irrespective of the test system used and the toxicity endpoint evaluated, thus extending the applicability of these test methods to strongly colored chemicals [18]. Another use of reconstructed epidermis is for the evaluation of skin corrosion where the time of contact of the product with the epidermis is shorter than for skin irritation [19].

As with eye irritation tests, Cosmetics Europe have developed a decision tree for skin irritation. They concluded that the good correlation between in vitro and in vivo skin irritation assays, together with the substantial in-house experience of these assays allows sufficient confidence in the outcomes of these assays such that in-house safety assessments on new products can be made without the use of animal testing. A decision tree for hazard assessment and labelling, using a weight of evidence (WoE) approach, involves a step-wise evaluation of firstly, physicochemical characteristics, (Q)SAR and existing data, to identify and rule out corrosive chemicals for further testing; secondly, an in vitro corrosivity test; and, finally, an in vitro irritation test to distinguish between irritants and non-irritants. In conclusion, evaluation of the skin irritation potential of new chemicals for use in cosmetics can be confidently accomplished using only alternative methods [20].

*3.4. Skin Sensitization*

Skin sensitization is one of the principal concerns related to the use of cosmetics and is usually associated with fragrances, preservatives and hair dyes [21–24].

Most studies have been done in vivo and a lower percentage on humans using the patch test method, to evaluate potential skin irritation caused by cosmetic ingredients (Figure 3). There has been a slight increase in the percentage of studies performed using the patch test in recent years.

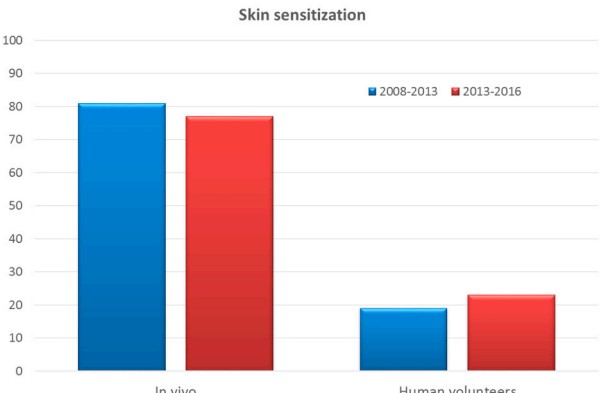

**Figure 3.** Evolution of the percentage of studies performed in vivo and, in human volunteers to evaluate potential skin sensitization of cosmetic ingredients—number of studies 87 (2008–2013) and 48 (2013–2016).

The accepted animal testing methods for skin sensitization potential assessment include the Mouse Local Lymph Node Assay (LLNA) and its non-radioactive modifications (LLNA-DA and the LLNA-BrdU ELISA) [25], the Guinea Pig Maximisation Test by Magnusson and Kligman (GPMT) and the Buehler occluded patch test in the guinea pig [26]. The most frequently used method was LLNA, but guinea pigs were also used in different studies.

The Local Lymph Node assay is considered a reduction and refinement method compared to traditional guinea pigs tests since it provides advantages in terms of animal welfare and was used until March 2013, before the full ban on animal use. Since this date, it has no longer been used to evaluate cosmetic ingredients.

No studies on skin sensitization were performed in vitro, despite the fact that there are some validated and OECD accepted methods, but these have only recently been accepted and the studies considered in this paper were performed before 2013. The acceptance of these methods suggests that, in subsequent years, in vitro studies of skin sensitization will be presented to the SCCS for evaluation.

One of the last in vitro methods to be accepted by the OECD is the In Vitro Skin Sensitisation or human cell activation test (h-CLAT), which is based on activation of the human monocytic leukaemia cell line THP-1 [27] as developed previously [28]. Other were accepted in 2015, including Test 442C, the In Chemico Skin Sensitisation: Direct Peptide Reactivity Assay (DPRA) [29], which was developed some years ago [30,31] and Test 442D: In Vitro Skin Sensitisation: ARE-Nrf2 Luciferase Test Method [32]. This last method was proposed to demonstrate the second event that takes place in the sensitisation mechanism occurring in the keratinocytes and includes inflammatory responses as well as gene expression associated with specific cell signaling pathways such as the antioxidant/electrophile response element [33].

A recent paper has reviewed the different opportunity of the in vitro methods for the assessment of contact sensitizers [34]

*3.5. Dermal Absorption*

Dermal absorption is a well established in vitro method that is described in the OECD guidelines and for which the SCCS have produced a special memorandum describing the requisites for the evaluation of cosmetic ingredients, such as minimal number of replicates, and the amount of substance applied on the Franz cells used in this test [35]. Despite the existence of an in vitro protocol, over the period

being considered, some studies were performed on animals and human volunteers, with an increase in the number of these studies and a decrease in the number of in vitro studies (Figure 4).

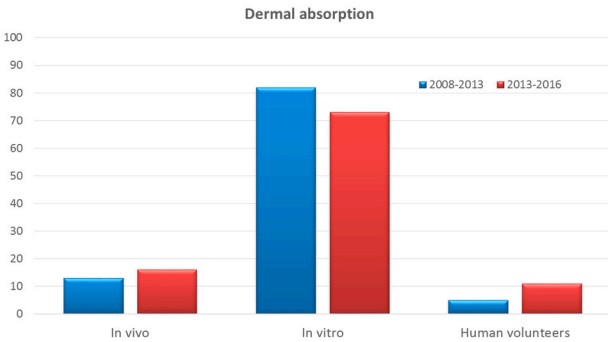

**Figure 4.** Evolution of the percentage of studies performed in vivo, in vitro and in human volunteers to evaluated dermal absorption of cosmetic ingredients—number of studies 86 (2008–2013) and 44 (2013–2016).

The in vivo studies were done in rats, and only one used rabbits, accounting for 16% of the total studies of dermal absorption. The SCCS recommends the use of human or pig for the in vitro method, with human skin being preferred if available. Human skin from surgeries was the most used tissue.

One study used the chorioallantoic membrane as an alternative to Franz cells. This method was described some years ago and the authors found the membrane to be similar to the buccal mucosa in terms of permeation profile and permeability coefficient. For this reason, it is not appropriate for studying dermal absorption [36].

Some dossiers did not include studies of dermal absorption, in which case an absorption of 100% was assumed per default, as is indicated in the notes of guidance of the SCCS [37].

None of the studies used cultured or reconstructed human skin models because they have been demonstrated to be inadequate due to their insufficient barrier function [38]. However, a recent paper described a 3D model of congenital ichthyosis, representing severe epidermal barrier function defects, which could be used to study dermal penetration [39].

Dermal absorption is an important factor in assessing the systemic toxicity of cosmetic ingredients. Compounds with low dermal bioavailability do not need to be evaluated for repeated dose toxicity, for which in vitro methods do not yet exist [40].

*3.6. Genotoxicity*

Studies of genotoxicity/mutagenicity in the dossiers presented to the SCCS showed an increase in the percentage of in vitro methods in the period 2013–2016 compared to the previous period evaluated (Figure 5).

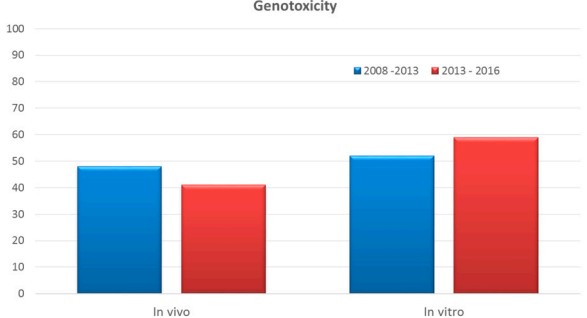

**Figure 5.** Evolution of the percentage of studies performed in vivo and in vitro to evaluate potential genotoxicity of cosmetic ingredients—number of studies 64 (2008–2013) and 175 (2013–2016).

There are a number of validated and accepted methods for studying genotoxicity. In the case of cosmetics, the SCCS has made a recommendation about which in vitro test should be used with the exception of special cases for which the Ames test is not suitable, the SCCS recommends two assays for the base level testing of cosmetic substances: the Bacterial Reverse Mutation Test or Ames test [41] that test for gene mutations and the In Vitro Micronucleus Test [42] that test for both structural (clastogenicity) and numerical (aneugenicity) chromosome aberrations [43].

If using both methods, there are four potential scenarios. If both tests are negative, then there is no mutagenic potential and further testing is not necessary. If the Ames test is negative and the in vitro micronucleus test is positive, then the substance may be considered an in vitro mutagen and further testing may be essential to clarify the clastogenic potential of the substance. In this case, either the Comet assay in mammalian cells [44] or in the 3D-reconstructed human skin model [45], or the micronucleus test in the 3D-reconstructed human skin model should be considered [46].

If the Ames test is positive and the micronucleous test is negative, the substance may be considered an in vitro mutagen. Further testing can be used such as an in vitro mammalian gene mutation test [47]. If the results from both tests are clearly positive in adequately performed tests, it is very likely that the substance has mutagenic potential and further confirmatory tests are not necessary.

The SCCS recommends that both tests should be done, but looking at the different methods used in the studies included in the dossiers, this is not always the case, and a considerable number of in vivo studies were presented for safety evaluation of the different cosmetic ingredients because they correspond to old studies performed before the ban.

However, as is shown in Figure 6, there has been an increase in the number of in vitro methods used in studies. Among these studies, 54% only performed the Ames test, 7% only performed the micronucleus test and 39% performed both as is shown in Figure 6. More studies used both tests after the publication of the SCCS recommendation. Two studies in which the Ames test was positive and the micronucleous test was negative subsequently performed the Comet assay in a 3D epidermis model following this recommendation.

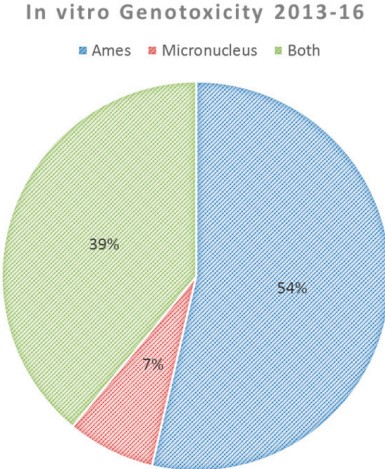

**Figure 6.** Percentage of the different in vitro studies of genotoxicity of cosmetic ingredients performed in the period 2013–2016.

A recent retrospective analysis of the mutagenicity/genotoxicity data of cosmetic ingredients has revealed that the in vitro test showed a low specificity for cosmetic ingredients. The different assays generated a high percentage of misleading positive results. As concluded by the authors of this study, there is a need of better regulatory strategies for cosmetic ingredients [48].

### 3.7. Carcinogenicity

Not all the dossiers included studies of carcinogenicity. Among the dossiers evaluated, the studies of carcinogenesis were generally done in vivo, using rats, mice or rabbit via oral or dermal administration.

As can be seen in Figure 7, the percentage of studies performed in vitro was very low with a slight decrease between the two periods under consideration. The methods used for the in vivo studies did not follow any guidelines, although there is an OECD guideline that was adopted in 1981 and was recently reviewed [49].

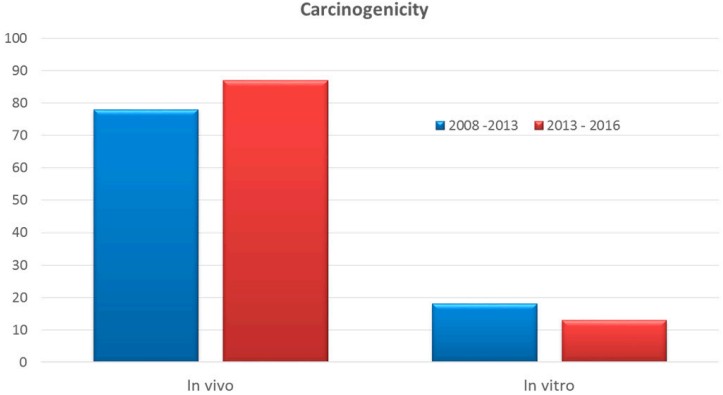

**Figure 7.** Evolution of the percentage of studies performed in vivo and in vitro to evaluate potential carcinogenicity of cosmetic ingredients—number of studies 26 (2008–2013) and 16 (2013–2016)

The in vitro studies in the dossiers under evaluation used the malignant transformation of C31-1-mouse M2-fibroblasts assay, similar to in vitro Bhas 42 cell transformation assay. The latter cell line was established by the transfection of the v-Ha-ras oncogene into the BALB/c 3T3 A31- 1-1 cell line and represents one of the best-known cell transformation assays for screening single chemicals or complex mixtures for carcinogenicity prediction [50]. The method has been validated [51] but has not yet been accepted and only a draft of the OECD guideline is available [52].

A guidance document on the in vitro Bhas 42 cell transformation assay can be downloaded at: http://www.oecd.org/officialdocuments/publicdisplaydocumentpdf/?cote=ENV/JM/MONO(2016)1&docLanguage=En [53].

Another in vitro test for carcinogenicity that was not used in the documents of the dossiers evaluated between 2009 and 2013, but was present in previous dossiers, is the Syrian hamster embryo cell (SHE) assay. This test has been used, almost since its initial description, as an in vitro test for determining the potential carcinogenicity of substances [54] and has been validated in various studies [55].

### 3.8. Toxicokinetic Studies

Toxicokinetic studies consider different processes such as absorption, distribution, metabolism and excretion (ADME). Many in vitro systems have their pitfalls, especially with respect to insufficient reflection of the integrated in vivo physiological ADME conditions and the lack of fully validated assays [56].

Comparison of the two periods evaluated in this paper revealed that the number of in vivo studies has increased in recent years, but with a reduction in studies on human volunteers, and that there has been no change in the use of in vitro studies (Figure 8). The small number of studies performed in vitro can be attributed to the lack of validated methods on toxicokinetics.

The in vivo toxicokinetic studies were performed in different animals by different routes, mainly oral and dermal. Few studies were performed on human volunteers after topical application of the

formulations. A small number of in vitro studies were conducted to demonstrate intestinal absorption using CaCo-2 cells [57,58] and others to determine metabolism in the skin using HaCaT keratinocytes, which have recently been demonstrated to be a model cell for the study of metabolomics effect of nanoparticles [59]. None of the studies used reconstructed epidermis, despite the fact that this has been demonstrated as a good strategy to study metabolism in vitro [60].

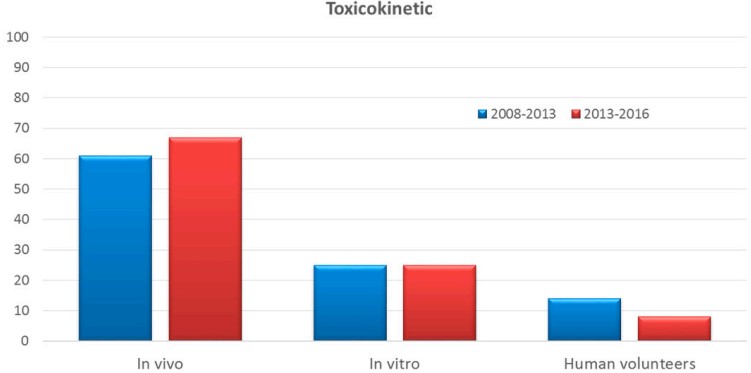

**Figure 8.** Evolution of the percentage of studies performed in vivo, in vitro and in human volunteers to evaluated toxicokinetic of cosmetic ingredients—number of studies 56 (2008–2013) and 36 (2013–2016).

The liver is the principal organ responsible for metabolism following absorption of a substance and its arrival in blood circulation. Isolated hepatocytes were thus used as it has been demonstrated that appropriate toxicokinetic information can be generated based solely on in vitro data, with the resulting data being in the same order of magnitude as those published for human volunteers [61].

*3.9. Phototoxicity*

Studies of phototoxicity were performed in the case of products that are particularly designed for exposure to sun radiation, such as UV filters but also for other products such as some hair dyes, preservatives, etc. Only 13 ingredients, which represents 32% of the products evaluated were studied for phototoxicity. Compared to the previous period of evaluation, the more recent period showed a decrease in the number of studies performed in vitro (Figure 9).

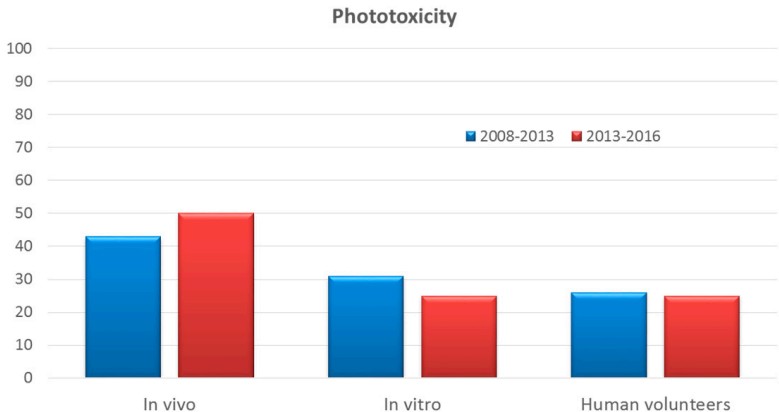

**Figure 9.** Evolution of the percentage of studies performed in vivo, in vitro and in human volunteers to evaluate potential phototoxicity of cosmetic ingredients—number of studies 35 (2008–2013) and 20 (2013–2016).

The in vitro method used to evaluate phototoxicity is based on the uptake of neutral red by the fibroblast cell line 3T3. This method has been validated and accepted in the OECD guidelines [62].

## 4. Conclusions

Based on the European ban on the use of animals to evaluate the safety of cosmetic ingredients that entered into force in March 2013, toxicological studies should be performed in vitro. However, the safety evaluation can be based on in vivo studies performed before the European ban on the use of animals. Comparison of the evaluations of the different cosmetic ingredients performed by the SCCS during the period 2013–2016 and 2009–2013 revealed a slight increase in the use of some in vitro methods but not as much as expected. The reason for this is that many of the ingredients evaluated were not new, and so the studies were based on tests conducted before the ban. The SCCS evaluates certain cosmetic ingredients more than once and then old toxicological data are described, together with the new data and for this reason there is a slight increase in the percentage of in vitro studies. Despite the old studies are considered with the new ones, the more relevant observation is the use of different in vitro methods.

Dossiers are a compilation of existing data and only if a test is not of enough quality or missing is the new data required by the SCCS. New data corresponding to studies on animals are accepted if they are performed before 2013. However, some in vitro studies were performed after 2013 and this explains the slight increase in the use of alternative methodologies. The recent acceptance of different alternative methods for skin sensitization should result in an increase in the number of such studies in the dossiers presented by the industry for evaluation in the next few years. A great effort is done by researchers to develop new in vitro methods and is done by industry to implement these new methods for the evaluation of cosmetic ingredients. After the ban, new ingredients should be studied without animals using in vitro, in chemico or by computational tools such as QSAR only.

**Author Contributions:** Maria Pilar Vinardell and Montserrat Mitjans contributed equally in the accomplishment of the study and in the writing of the manuscript.

**Conflicts of Interest:** The authors declare no conflict of interest.

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
