# Peer review of "Alternative Methods to Animal Testing for the Safety Evaluation of Cosmetic Ingredients: An Overview"

_cosmetics, doi:10.3390/cosmetics4030030_

Round 1

Reviewer 1 Report

Dear Authors,

I like your paper because it is interesting the way  you consider the comparaison between safety tests performed by the in vivo methods and in vitro method.

Hovewer I suggest to modify the title of the paper. In this form it seems you want to perform some  in vitro test instead to make an overview on test performed in the last years.

Author Response

According to the comments of the reviewer I have added  at the end of the tittle"An overview"

Reviewer 2 Report

This manuscript provides an overview of the use of non-animal approaches for the safety evaluation of cosmetic ingredients in the EU. It is a follow-up of a previous inventory. The manuscript is easy to ready and informative. I have only a few comments. 

Page 1 lines 20-21. Cosmetic Industry is responsible for the safety of cosmetic ingredients. SCCS is mandated by the EC to provide a scientific opinion on the dossier submitted by industry. Please revise this accordingly.

The inventory is based on SCCS opinions published from 2013-2016. The SCCS evaluates certain cosmetic ingredients more than once, for example because industry provides them with new data. It would be good to distinghuish first submission from subsequent submissions. In subsequent submission often the old tox data are described, together with the new data. This means that old animal studies will be counted again in the second inventory and is not representative of the use of non-animal methods for newly evaluated cosmetic ingredients. 

Page 7, the default dermal absorption is 50% according to the SCCS Notes of Guidance and not 100%. 

It would be good to mention to know where the percentages in each Figure are based on. Please add the total number of substances in the footnote of each Figure. Only percentages are not informative to the reader. 

The last sentences of the Discussion are somewhat strange. Not only industry, but several universities and institutes work on alternatives to replace animal testing. So please make this sentence more general. Second, the last sentence suggests that only in vitro assays will be used in the future. But computational tools (QSARs for example) and in chemico assays (think of the DPRA assay) will be used as well. A more general statement would be better. In this case wording like testing strategies, defined approaches or IATA would be better, because a single in vitro assay will never replace an animal test. 

Author Response

This manuscript provides an overview of the use of non-animal approaches for the safety evaluation of cosmetic ingredients in the EU. It is a follow-up of a previous inventory. The manuscript is easy to ready and informative. I have only a few comments. 

Page 1 lines 20-21. Cosmetic Industry is responsible for the safety of cosmetic ingredients. SCCS is mandated by the EC to provide a scientific opinion on the dossier submitted by industry. Please revise this accordingly.

Answer: It has been revised accordingly and changed

The inventory is based on SCCS opinions published from 2013-2016. The SCCS evaluates certain cosmetic ingredients more than once, for example because industry provides them with new data. It would be good to distinghuish first submission from subsequent submissions. In subsequent submission often the old tox data are described, together with the new data. This means that old animal studies will be counted again in the second inventory and is not representative of the use of non-animal methods for newly evaluated cosmetic ingredients.

Answer: a comment to clarify this points has been included  

Page 7, the default dermal absorption is 50% according to the SCCS Notes of Guidance and not 100%. 

Answer: changed

It would be good to mention to know where the percentages in each Figure are based on. Please add the total number of substances in the footnote of each Figure. Only percentages are not informative to the reader. 

Answer: The total number of studies performed have been included. In more cases the same substance has been studied with different methods, then the number of studies is higher than the number of substances evaluated

The last sentences of the Discussion are somewhat strange. Not only industry, but several universities and institutes work on alternatives to replace animal testing. So please make this sentence more general. Second, the last sentence suggests that only in vitro assays will be used in the future. But computational tools (QSARs for example) and in chemico assays (think of the DPRA assay) will be used as well. A more general statement would be better. In this case wording like testing strategies, defined approaches or IATA would be better, because a single in vitro assay will never replace an animal test. 

Answer: it has been changed accordingly

Round 2

Reviewer 2 Report

I have looked at the revised manuscript. The authors have revised the manuscript according to my comments. I would recommend publication now.